# Preparation of Microcellular High-Density Polyethylene with Thermal Expandable Microspheres

**DOI:** 10.3390/polym17131773

**Published:** 2025-06-26

**Authors:** Guo-Shun Chen, Xue-Kun Li, Wei-Cheng Yang

**Affiliations:** 1State Key Laboratory of Polyolefins and Catalysis, Shanghai 200062, China; chenguoshun0329@163.com; 2Guangxi Key Laboratory of Green Chemical Materials and Safety Technology, Qingzhou 535000, China; 3Shanghai Research Institute of Chemical Industry Co., Ltd., Shanghai 200062, China; 4Shanghai Key Laboratory of Catalysis Technology for Polyolefins, Shanghai 200062, China

**Keywords:** injection molding, thermal expansion microspheres, foamed high-density polyethylene

## Abstract

The microstructure and mechanical properties of foamed high-density polyethylene (HDPE) prepared with thermal expandable microspheres (TEMs) by an injection molding method were investigated, especially for the effect of different injection times, nozzle temperatures, and TEM contents. The results showed that it was beneficial to increase the expansion ratio in the HDPE of microspheres with a shorter injection time and higher nozzle temperature. However, the addition of TEMs reduced the crystallinity of the foamed HDPE, and the crystallinity decreases further with the increasing TEMs content, which led to the decrease in Young’s modulus and tensile strength. When the nozzle temperature reached 220 °C, the mechanical properties of the foamed HDPE diminished significantly due to the collapse of the internal cells. At the TEMs content of 1.5 wt.%, an injection time of 2.0 s, and a nozzle temperature of 210 °C, a foamed HDPE was prepared with a cell size of 89.72 μm and a cell density of 4.39 × 10^8^ cells/cm^3^. This foam exhibited a reduction density of 5.75%, a tensile strength of 22.6 MPa, and a Yang’s modulus of 1172.3 MPa, demonstrating excellent overall mechanical properties.

## 1. Introduction

In the 21st century, the automotive manufacturing industry is exhibiting development trends characterized by electrification, connectivity, intelligence, and sharing. Within this context, automotive lightweighting stands as one of the critical strategic measures to achieve the modernization of the automotive industry. Automotive lightweighting involves reducing the curb weight of a vehicle while ensuring its strength and safety performance, thereby enhancing the power performance and achieving energy conservation and emission reduction objectives. Furthermore, automotive lightweighting contributes to active safety by effectively improving operational stability and reducing braking distance [1,2]. A 10% reduction in vehicle weight could lead to a 5% decrease in braking distance and a 6% reduction in steering force [3].

The approaches to realizing automotive lightweighting encompass material application lightweighting [4], structural design lightweighting [5], and manufacturing process lightweighting [6]. Among these, material lightweighting represents the most significant lightweighting technology. Key materials for automotive lightweighting include high-strength steel, aluminum alloys, magnesium alloys, titanium alloys, and other metallic materials, as well as engineering plastics, non-metallic composite materials, and foamed engineering plastics.

High-density polyethylene (HDPE), due to its exceptional toughness, rigidity, chemical resistance, wear resistance, low water and moisture absorption, and high temperature tolerance, has become an essential component in the production of automotive non-metallic parts. The foams based on HDPE inherit these properties. Compared to HDPE, although low-density polyethylene (LDPE) and linear low-density polyethylene (LLDPE) exhibit better foamability, their mechanical properties are inadequate to meet the structural requirements of HDPE in the automotive industry. Engineering plastics with superior strength, such as PA, are more expensive and thus less frequently utilized compared to HDPE in automotive applications. Consequently, lightweight HDPE demonstrates greater potential economic value by reducing the vehicle weight and energy consumption, while also offering innovative insights into the development of high-performance lightweight composite materials, such as anisotropic foam structures.

The cell structure of foamed polymers significantly influences the properties. Wang [7] prepared an isotactic polypropylene (iPP)/HDPE composite foam with excellent comprehensive mechanical properties through co-injection foaming. However, the majority of cell sizes exceed 100 μm, which resulted from the chemical blowing agent. In comparison to macrocellular foamed polymers, microcellular foamed polymers offer several advantages, including a reduced cycle time, lower stress on the machining die, enhanced dimensional stability of components, and minimal degradation of the mechanical properties of polymers [8,9,10,11].

Supercritical fluids are commercially used as foaming agents in the production of lightweight injection-molded components through the injection molding process. The expansion of the supercritical fluids inside of the polymer melt creates a uniform packing effect and prevents the material from shrinking during cooling, leading to the excellent dimensional stability of the molded samples [12,13]. However, due to the high investment in equipment and the high pressure during the processing, the scale of production is somewhat limited. Additionally, the effects of the induced crystallization and other factors lead to the stretching and collapse of some cells on the surface of the samples, resulting in the presence of vortices and silver stripes on the surface, which affects the appearance of the samples and makes them unsuitable for products that require a high transparency and an extremely high surface quality. Although several methods, such as gas counter-pressure, vapor polishing, hot/cold mold cycling [14], expanding mold, insulating film coating [15], mold surface treatments [16], and co-injection molding [17], could be used to enhance the surface quality of microcellular injection-molded components, it may increase the complexity of the process and incur additional costs.

Thermal expansion microspheres (TEMs) are micro-scale spheres and a new type of physical foaming agent, featuring a core–shell structure with an acrylate copolymer as the shell and hydrocarbon as the core [18]. When heated, the shell softens, the low-boiling-point hydrocarbon in the core vaporizes, and the microsphere expands in volume. Thermal expansion microspheres are employed as a blowing agent to fabricate microcellular injection-molded samples with relatively uniform micropores and an improved surface quality [19]. There are no cells stretching and collapsing that would cause vortices and silver stripes on the surface, resulting from the sealed gas expansion within the expandable microspheres structure. In addition, it can eliminate additional equipment investments because no additional license fees for gassing systems (e.g., *Texel’s MuCell^®^ process*) and no machine changes are needed. The application of TEMs in the foamed materials such as LLDPE and polylactic acid (PLA) has been documented in the literature. However, the utilization of TEMs in foamed HDPE remains underexplored.

Therefore, TEMs were utilized as a blowing agent to fabricate foamed injection-molded high-density polyethylene (HDPE) samples in this paper. And the effects of the injection molding process and the content of TEMs on the cell structure and mechanical properties of injection-molded samples were investigated through techniques such as SEM, DSC, and mechanical property testing. It holds significant reference value for the molding and preparation of foamed HDPE in the automotive industry.

## 2. Materials and Methods

### 2.1. Materials

The materials used in this study include a commercial-grade high-density polyethylene (HDPE) homopolymer from SINOPEC-SK Wuhan Petrochemical Co., Ltd. (Wuhan, Hubei, China). with a melt index of 0.2 g/10 min. Thermal expansion microspheres (TEMs)—F190D from Matsumoto Yushi-Seiyaku Co., Ltd. (Hyogo Prefecture, Osaka, Japan). and PG31 and PG42 from Dongjin Semichem Co., Ltd. (Seoul, South Korea)—were employed as the blowing agent additive. The TEMs and HDPE were mixed using a three-dimensional mixer, before using UN120SJ injection molding machine (YIZUMI, Foshan, Guangdong, China) to manufacture the tensile bars (ASTM D638-08 Type I standard [20]), bending bars (ASTM D638-08 Type I standard), and impacting bars (ASTM D638-08 Type I standard).

### 2.2. Expansion Behavior Characterization of TEMs

The expansion temperature and the expanding behaviors of TEMs were characterized by thermomechanical analysis (TMA-4000, PerkinElmer, Waltham, MA, USA). A sample (25 µg) was put into a cup and heated from ambient temperature to 250 °C at a rate of 5 °C·min^−1^. The temperature at which the expansion started is called *T*_start_ (measured by TMA), whereas the temperature at which the maximum expansion was reached is called *T*_max_ (measured by TMA). The change in dimension at which the maximum expansion was reached is called *D*_max_ (measured by the TMA).

### 2.3. Thermal Analysis

The heat change during the expansion process of the microspheres was analyzed using differential scanning calorimetry (DSC 3^+^, Mettler Toledo, Zurich, Switzerland) under a high-purity nitrogen atmosphere. Samples, with weights ranging from 3 to 5 mg, underwent heating from 30 to 250 °C at a rate of 10 °C/min.

The melting point and crystallinity of unfoamed HDPE and foamed HDPE were examined employing a DSC. Samples, with weights ranging from 5 to 10 mg, underwent heating from 30 to 220 °C at a rate of 10 °C/min and then cooling down to 30 °C at the same rate in a nitrogen atmosphere. The crystallinity of samples (*χ*_HDPE_) was obtained using the following equations:(1)$$\chi_{HDPE}=\left(\frac{{∆H}_{m,\;HDPE}}{{∆H}_{m,\;HDPE}^{0}}\right)\times 100\%$$
where Δ*H*_m,HDPE_ was derived from the DSC thermograms of the foamed HDPE. The fusion enthalpy Δ*H*^0^_m, HDPE_ of 100% crystalline polymer is 293 J/g for HDPE [21].

### 2.4. Microstructure Characterization of TEMs and Foamed HDPE

The microstructure was observed by scanning electron microscope (SEM-Merlin compact, Zeiss, Oberkochen, Baden-Wurttemberg, Germany). For TEMs, samples were subjected to heating at 180 °C, 200 °C, and 220 °C for 5 min, respectively, followed by gold sputtering for SEM characterization. For foamed HDPE, all the samples were freeze-fractured in liquid nitrogen and then cut by a sharp blade. The fractured surfaces were sputter-coated with gold before testing. The average cell size and density were analyzed using an image analysis tool (Image J 1.52a), and more than 100 cells were measured. The cell density (*N*) is calculated by following Equation [22]:(2)$$N={\left(\frac{n}{A}\right)}^{3/2}\varphi$$(3)$$\varphi =\rho_{0}/\rho_{f}$$
where *n* represents the cell number in the SEM micrograph, *A* denotes the micrograph area, *ρ*_0_ is the density of the pre-foam, and *ρ_f_* signifies the density of the post-foam sample measured via the Archimedes method.

### 2.5. Density Characterization

The density data of the foamed samples were obtained by the water displacement method based on Archimedes’ principle using a density determination kit for the analytical balance (Mettler Toledo ML204, Zurich, Switzerland). At least three measurements were carried out for each sample.

### 2.6. Mechanical Properties

Mechanical test samples were obtained directly by injection molding following the sizes specified in ASTM testing standards. The tensile tests experiments were conducted at room temperature, employing a tensile tester (UTM-1432, JJ testing Instrument Co., Ltd., Yancheng, Jiangsu, China). Each condition was tested on five samples, and average values were computed. The crosshead speed was set at 50 mm/min for unfoamed and foamed HDPE. All tests were carried out according to the ASTM D638-08 standard. The apparent stress and strain were calculated based on the original tensile bar’s cross-sectional area and length, respectively.

For evaluating the impact strength variation, ASTM D256 standard [23] type bars were fabricated, incorporating a 2 mm V-shaped groove at the center of the sample. Impact strength testing experiments were conducted using an impact tester (XJJD-5, JJ testing Instrument Co., Ltd., China). Unfoamed and foamed tensile bars were molded at the processing conditions shown in Table 1.

## 3. Results and Discussion

### 3.1. The Thermal Expansion Behavior of the TEMs

To identify TEMs with appropriate expansion temperatures for HDPE, their expansion curves were assessed using the TMA and are shown in Figure 1. It could be observed that all curves expressed a typical trend, which could be divided into four phases. Phase I: When the temperature was under the *T*_start_, even though the temperature had exceeded the vaporization temperature of the alkanes inside the TEMs, the shell polymer of the TEMs had not yet reached the softening temperature, the microspheres were unexpanded, and the displacement was almost constant. Phase II: As the temperature continued to increase, the mobility of the polymer’s molecular chains enhanced, leading to a gradual softening of the shell layer. Under the influence of the internal gas pressure, some microspheres began to expand, causing the TMA curve to start rising. Herein, this temperature was designated as *T*_start_, representing the threshold temperature required for the microspheres to initiate expansion. When the dimension value reached its maximum, it signified that the majority of microspheres have essentially completed their expansion process, and the corresponding temperature was defined as *T*_max_. Phase III: Beyond the *T*_max_, some polymers had reached the melting point, and the shell layer was too soft to bind the internal gas, causing the microspheres to rupture and collapse. The TMA curve began to show a downward trend. This was attributed to the fact that some microspheres were coated with too many alkanes during the polymerization process or that the molecular weight of the shell layer polymer was relatively low and its mechanical properties were relatively weak. Phase IV: With the further increase in the temperature, the displacement did not decrease further, which indicated that not all of the microspheres had broken, but only some of the microspheres had been destroyed. Although the temperature was still rising and the shell polymers had all reached the viscoelastic state, they benefited from the stacking effect, the expanded microsphere could be maintained until the polymers started to decompose without the external force compression. It is worth noting that the *T*_start_ of PG42, PG31, and F190D was 121 °C, 144 °C, and 167 °C, while the *T*_max_ was 171 °C, 202 °C, and 245 °C, respectively. The different values obtained for the expansion start, maximum expansion, and maximum processing temperature can be explained by the varying structure and composition of the microspheres. For instance, molecular structures such as -COOH groups with strong electrostatic interactions and hydrogen-bonding forces were suitable for the high heat resistance [24], and increasing the crosslinking density of the polymer shell had similar effects [25].

Furthermore, as illustrated by the DSC curve in Figure 2, with the increasing temperature, distinct endothermic peaks emerged on the DSC curves. The endothermic peak temperature was highest for F190D, followed by PG31, and was lowest for PG42, which aligned with the findings from the TMA curve analysis. Notably, the onset temperature of the endothermic peak was significantly lower than the *T*_start_ observed in the TMA results. This suggested that prior to the expansion of the TEMs, the alkanes within the core material undergo endothermic vaporization. However, at this stage, the shell polymer had not yet reached its viscoelastic state, maintaining a relatively high mechanical strength. Consequently, the outward expansion pressure generated by the vaporization remained lower than the mechanical strength of the polymer shell, preventing any volumetric changes in the microspheres. As the temperature continued to rise, the shell polymer entered a viscoelastic state, leading to a reduction in its mechanical strength while enhancing its elasticity. At this point, the outward expansion pressure caused by the alkane vaporization exceeded the mechanical strength of the shell polymer, initiating the outward expansion of the microspheres.

Although the *D*_max_ value of F190D was the largest one (*D*_max_ = 3.0983 mm, showed in Figure 1), which indicates the highest expansion rate, its expansion temperature range (167–245 °C) could not match the processing temperature (160–210 °C) of HDPE, leading to a low expansion rate. Similarly, the expansion temperature (121–171 °C) of PG41 was excessively low, causing the temperature to exceed the maximum expansion temperature within the HDPE processing temperature range, which results in the rupture of the microspheres and a reduced expansion rate. It can be anticipated that, in comparison to PG41 and F190D, the expansion temperature of PG31 is more suitably aligned with the processing temperature of HDPE. Additionally, it can achieve a higher expansion ratio while maintaining an excellent closed-cell structure.

As illustrated in Figure 3, for PG31 the average diameter prior to expansion was approximately 30–40 μm (Figure 3a). As the temperature increased, the average diameter began to gradually increase and reached a peak value of around 140 μm at 200 °C (Figure 3c). When the temperature rose to 220 °C, some alkanes escaped from the polymer shell, causing the microspheres to shrink or even rupture (Figure 3d). This observation aligned with the findings presented in Figure 1.

The critical factor in the production of homogeneous and closed-cell structural foams incorporating microspheres is the maximum temperature at which the microspheres can remain suspended without collapsing (the rupture of the shell) or shrinking. Therefore, PG31 was selected as the blowing agent for the preparation of the microcellular HDPE.

### 3.2. Effect of Injection Time on Mechanical Properties of Foamed HDPE

The microcellular structure can be affected by the injection volume, injection speed, and mold temperature. An intermediate injection speed and low mold temperature were beneficial for the homogeneous microcellular structure [26]. Thus, the mold temperature was 20 °C and the injection speed was 133mm/s constantly. In addition, as can be observed from Figure 1, the expansion temperature of PG31 ranged from 144 °C to 202 °C. To prevent the rupture of the microsphere shell caused by excessively high temperatures within the screw, which could negatively impact the expansion rate, a nozzle temperature of 190 °C was selected. Furthermore, the temperatures in all other intervals are maintained below 180 °C to ensure the structural integrity of the microsphere shell.

Table 2 summarizes the values of Young’s modulus, the ultimate tensile strength, and the strain at break for five samples tested according to the ASTM D638-03 standard. The density and reduction in density were also calculated for five samples. Reducing the resin injection volume by shortening the injection time—leaving more space inside the mold to facilitate the expansion of the TEMs, which could cause the sample to be foamed sufficiently—resulted in larger cell sizes and lower densities. The mechanical properties of Young’s modulus, the strain at break, and the ultimate tensile strength diminished as the injection time decreased, while the reduction in density was decreased from 0.9451 g/cm^3^ to 0.9066 g/cm^3^. In particular, Young’s modulus and the strain at break decreased sharply with the decrease in the injection time. This is likely associated with the growing cell size in the HDPE matrix [27].

The representative stress–strain curves of foamed samples with different injection times were featured in Figure 4. The stress–strain curves of HDPE samples can be categorized into two distinct stages. In the initial phase of the stress–strain curve (strain less than 10%), the material exhibits a Linear Elastic Region, where the stress increases linearly with the strain. During this stage, the sample undergoes uniform stretching in accordance with Hooke’s law, and the slope of the curve represents Young’s modulus. Upon reaching the maximum stress value (yield stress), the cross-sectional area of the sample becomes abruptly uneven, forming one or more “necking” regions, thereby transitioning into the second stage. In this stage, stress gradually decreases while the strain continues to increase until the sample fractures. With decreasing injection times, the density of the foamed HDPE diminished, leading to a reduction in material stiffness as evidenced by a decrease in the linear slope (Young’s modulus) during the linear elastic stage and a reduction in yield stress. However, the incorporation of TEMs introduced interfacial defects within the HDPE matrix. As tensile deformation progresses into the second stage, these defects further reduce the stress and strain of the foamed HDPE. Additionally, lower densities of foamed HDPE correspond to larger cell diameters, which result in reduced stress and strain values.

The variations in the bending modulus and bending strength as functions of the injection time are summarized in Figure 5. The bending modulus and bending strength of the foamed samples increased with the decrease in the injection time, resulting from the increasing of the cell sizes in foamed samples. Unlike traditional open-cell HDPE foam, when TEMs expand within HDPE, if the TEMs remain intact, they can form a closed-cell structure. Additionally, the shell of the TEMs exhibits a certain degree of compressive deformation resistance, which can effectively disperse stress under bending forces, preventing a localized stress concentration and thereby enhancing the material’s bending strength to some extent.

### 3.3. Effect of Nozzle Temperature on Cell Structure and Mechanical Properties of Foamed HDPE

To investigate the effects of the nozzle temperature on the microstructure of foamed samples, the cryogenically fractured cross-sectional morphologies of the injection-molded tensile bars were examined using scanning electron microscopy (SEM). The microstructure is shown in Figure 6, and the cell size and cell density are summarized in Table 3. The cell size increased from 59.53 μm to 127.44 μm, while the cell density decreased with the rising nozzle temperature. It is important to highlight that, in comparison to the expansion in air (shown in Figure 3), the microsphere diameter in the HDPE resin is considerably smaller prior to the microsphere rupture (before 200 °C). Conversely, the bubble diameter in foamed HDPE becomes significantly larger following the microsphere rupture (after 200 °C). Due to the inherent viscosity of the HDPE melt at high temperatures, it exerts a certain degree of external pressure on the expansion of microspheres, thereby restricting their volume expansion to a certain extent. Consequently, the diameter of microspheres after expanding within HDPE was smaller compared to those that expanded in air [28]. As the temperature increases, HDPE transitions from a glassy state to a melted state, thereby reducing the binding effect on the expansion behavior of TEMs. Simultaneously, the alkanes encapsulated by TEMs had transformed from a liquid to a gas, ultimately leading to the expansion of TEMs in the HDPE melt and the formation of the foam. However, it can be observed from Figure 6d that part of the cell size is even smaller than in Figure 6a, and the shell of the TEMs after the rupture and contraction can be seen in the hole. Although the higher temperature was conducive to improving the expansion efficiency of TEMs, once the temperature reached 220 °C, which was significantly higher than the *T*_max_ (202 °C, shown in Figure 1) of PG31, portions of the TEMs began to fracture and contract. The low-boiling alkanes coated in the microspheres escaped into the resin matrix, and a portion of the melt resin infiltrated the voids created following the rupture of the microspheres, leading to a decrease in the uniformity of cell sizes.

Figure 7 expresses the tensile strength variation in foamed samples with respect to the nozzle temperature. The mechanical properties of Young’s modulus, the strain at break, and the ultimate tensile strength (shown in Table 4) initially increased and subsequently decreased as the nozzle temperature rose. This is probably attributed to the crystallization behavior of the polymer. It is well established that polymer crystallization typically occurs within the temperature range between the melting point and the glass transition temperature, with the crystallization rate being governed by both nucleation and crystal growth kinetics. In high-density polyethylene (HDPE), the primary nucleation mechanism is homogeneous nucleation. Following nucleation, polymer chains extend radially from the nuclei in a process known as crystallization growth. Crystal growth is facilitated by molecular migration, with molecular mobility exhibiting a positive correlation with the temperature elevation. Consequently, as the nozzle temperature rises, the melt temperature also increases. The melt in the mold cavity requires extended cooling times, providing more time for crystal growth during cooling, thereby enhancing the polymer crystallinity and subsequently improving material mechanical properties. However, while high temperatures (220 °C) facilitate the crystallization of HDPE, the larger diameter and poor uniformity of the bubble holes formed by TEMs exerted a more significant negative influence on the mechanical properties of HDPE.

Figure 8 shows the bending and impact strength variation in foamed samples with respect to the nozzle temperature. The bending strength, bending modulus, and impact strength showed a trend of increasing first and then decreasing with the raising nozzle temperature. This can be inferred to the growing cell size in the foamed samples. In general, the cell morphology had a strong relationship with the mechanical properties. To improve the impact strength, the cell morphology had to consist of a well-developed uniform microcellular structure with a small cell size and a high cell-population density [29]. However, as shown in Figure 3d and Figure 6d, when the temperature reached 220 °C, a portion of TEM shells were broken and lost elasticity, resulting in the non-uniform microcellular structure in foamed HDPE. Thus, the dimensional stability of the foamed samples was affected when subjected to bending stress.

### 3.4. Effect of TEM Contents on Cell Structure and Mechanical Properties of Foamed HDPE

Figure 9 shows the cross-section morphology with different TEM contents, and the cell size and cell density are displayed in Table 3. There remained a trend of a decreasing cell size and an increasing cell density with the TEMs content. The average cell diameters are similar at approximately 89.72 μm, when the TEMs weight ratio was 1.5 wt.%, which corresponds to the insufficient expansion. It is widely recognized that melt strength significantly influences the expansion behavior of TEMs during molding. Specifically, a higher melt strength of the polymer matrix results in smaller cell sizes and higher foamed samples densities [30]. The high melt strength of the HDPE matrix restricted the complete expansion process of TEMs, resulting in a smaller density reduction (5.75%), while the same TEMs content in LDPE was 9.7% [30]. When the TEMs weight ratio is increased to 3.0 wt.% and 4.5 wt.%, the average cell diameter drops to 87.43 μm and 76.70 μm, respectively. In the confined cavity volume, the expansion volume of TEMs with higher contents is limited, resulting from the phenomenon of inflation competition and mutual inhibition.

There is no significant alteration in the external environmental pressure. Within the confined cavity volume, the high density of microspheres leads to phenomena akin to competitive expansion and mutual inhibition.

The representative stress–strain curves of unfoamed and foamed HDPE samples are featured in Figure 10. Table 5 tabulates the values of Young’s modulus, the ultimate strength, and the strain at break of tensile bars. As shown in Figure 10 and Table 5, the density of foamed samples reduced with the increasing TEMs content. The Young’s modulus and ultimate strength of unfoamed HDPE samples were higher than their foamed counterparts. By adding TEMs to the samples, the foamed structure with expanded TEMs affected the mechanical properties. With the increasing TEMs content, the Young’s modulus and ultimate strength of the foamed HDPE decreased sharply (from 1330.1 MPa and 22.8 MPa to 839.1 MPa and 15.8 MPa, respectively), while the strain at break was improved (from 37.06% to 41.39%). The degradation of mechanical properties was attributed to the presence of voids originating from TEM cells within the samples, which significantly diminished the effective cross-sectional area of the tensile test samples. At an ambient temperature, the amorphous regions of HDPE exhibit a rubbery state. As the content of TEMs increased, the crystallinity of the foamed HDPE decreased (as shown in Table 6), leading to a reduced tightness in the molecular chain arrangement and increased porosity. Upon impact, the activity space of the molecular chain segments expanded, resulting in an enhanced impact strength and an increased strain at break.

The variations in the bending modulus and bending strength as functions of injection time are summarized in Figure 11. The bending modulus and bending strength of the foamed samples increased first and then decreased with the increase in TEMs content (3.0 wt.% and 4.5 wt.% TEMs). When the TEMs content was 1.5 wt.%, the existence of TEMs was beneficial to improve the resistance of the material to the bending deformation. However, at the TEM contents of 3.0 wt.% and 4.5 wt.%, the excessive density of pores introduces more defects into the HDPE matrix, leading to stress concentration during tensile and bending processes, thereby reducing both the bending modulus and bending strength.

### 3.5. Effect of TEM Contents on Crystalline of Foamed HDPE

The results of DSC tests from the initial heating of the neat HDPE and foamed samples with different TEM contents are presented in Figure 12. The melt temperatures and the heat of fusion, derived from the DSC thermograms of the examined materials, are presented in Table 6. The area under the curve is representative of the heat of fusion, which is directly proportional to the polymer crystallinity.

**Table 6 polymers-17-01773-t006:** Melting and crystallization temperatures and heat of fusion for HDPE and foamed HDPE.

Material	UF-HDPE	F-HDPE-7	F-HDPE-9	F-HDPE-10
Melting onset temperature/°C	121.19	119.82	119.81	119.60
Melting peak temperature/°C	116.89	114.95	114.29	113.91
Melting end temperature/°C	90.39	85.40	85.21	85.52
Melting regime broadness/°C	30.80	34.42	34.60	34.08
Heat of fusion, Δ*H*_m,HDPE_ (J/g)	183.95	174.04	167.57	157.67
*χ*_HDPE_(%)	62.78	59.40	57.19	53.81

From the figure and table, it can be inferred that the degree of the crystallinity of foamed samples remarkably decreased from 62.78% to 53.81%, while the TEMs content increased from 0 to 4.5 wt.%. This can be attributed to the presence of TEMs which hindered the growth of polymer crystals, resulting in the lower degree of crystallinity. And the melting (peak) temperature of foamed samples decreases gradually from 116.89 °C to 113.91 °C with the increased TEMs content. The melting peak onset temperature and melting end point shift towards lower temperature when TEMs were added to HDPE matrix, while those remain nearly unchanged with the increased TEMs content. The difference between the melting end point and onset temperature could represent the broadness of a melting regime. These differences for UF-HDPE, F-HDPE-7, F-HDPE-9, and F-HDPE-10 are 30.80, 34.42, 34.60, and 34.08, respectively. Hence, a significant broadening in the melting regime was noticed for the foamed HDPE. And it can be stated that the size distribution of crystals is wider than the pure materials.

## 4. Conclusions

Foamed HDPE can be prepared by injection molding with TEMs as the foaming agent, and the process is similar to that of traditional chemical foaming agents. The microstructure and mechanical properties of the foamed HDPE injection-molded samples with thermal expandable microspheres (TEMs) were systematically investigated. Basically, the injection volume (injection time), nozzle temperature, and TEMs content can significantly affect the expansion ratio, and thus the cell size and morphology in the foamed samples are affected. It is beneficial to increase the expansion ratio of microspheres with a smaller injection volume and higher nozzle temperature, resulting in the density reduction in foamed samples. The cell size increased, while the cell density decreased with the rising injection volume and nozzle temperature. The addition of TEMs reduced the crystallinity of the foamed HDPE, leading to the decrease in Young’s modulus and the tensile strength. An appropriate TEMs content has a positive effect on the cell microstructure and weight reduction, whereas too high a concentration of microspheres adversely affects the tensile properties. By utilizing TEMs as a foaming agent, it is feasible to prepare closed-cell microcellular foamed polymers with superior mechanical properties at a relatively low cost, without the need for additional equipment investments. Consequently, TEMs demonstrate a promising and extensive application potential in the field of polymer foaming.

## Figures and Tables

**Figure 1 polymers-17-01773-f001:**
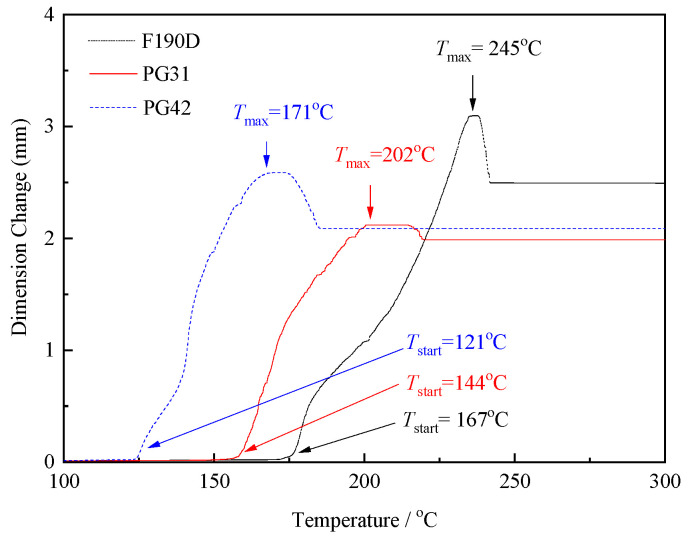
TMA curves with different TEMs.

**Figure 2 polymers-17-01773-f002:**
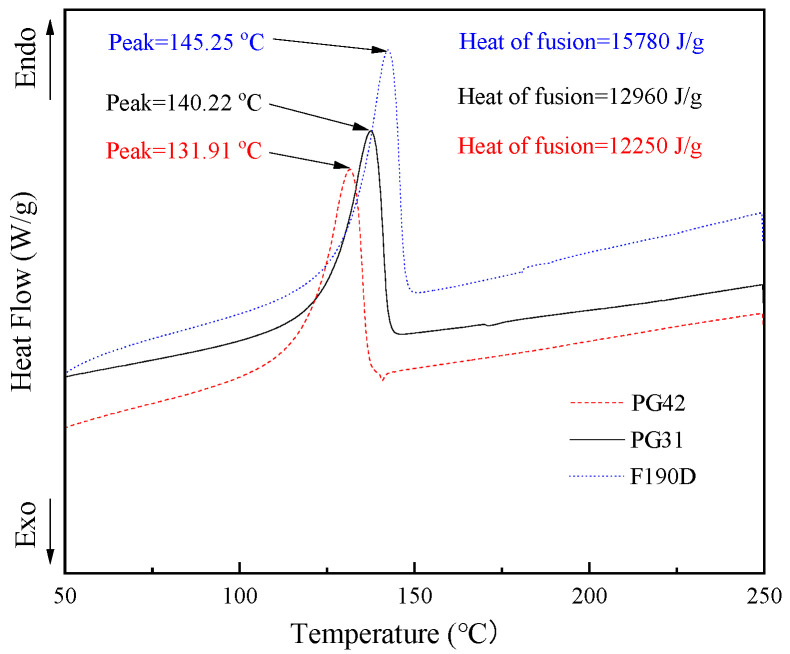
DSC curves with different TEMs.

**Figure 3 polymers-17-01773-f003:**
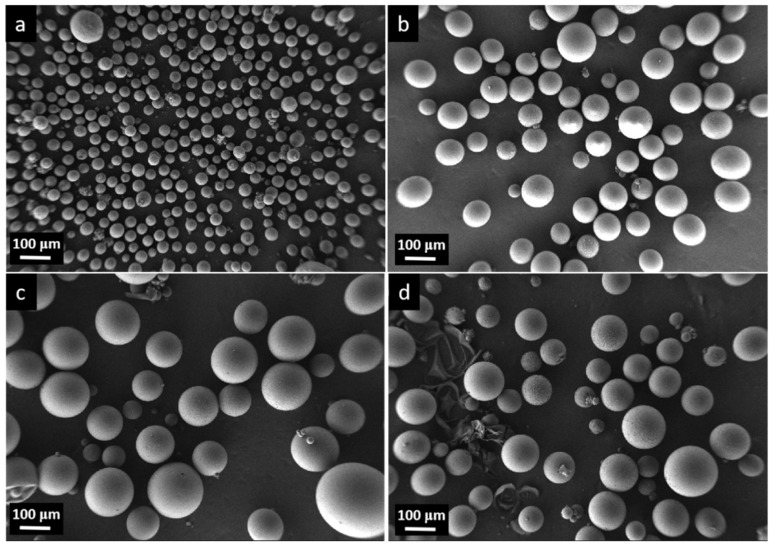
SEM images (Mag = 200) of PG31 with different temperatures: (**a**) before expansion, (**b**) expanded at 180 °C, (**c**) expanded at 200 °C, and (**d**) expanded at 220 °C.

**Figure 4 polymers-17-01773-f004:**
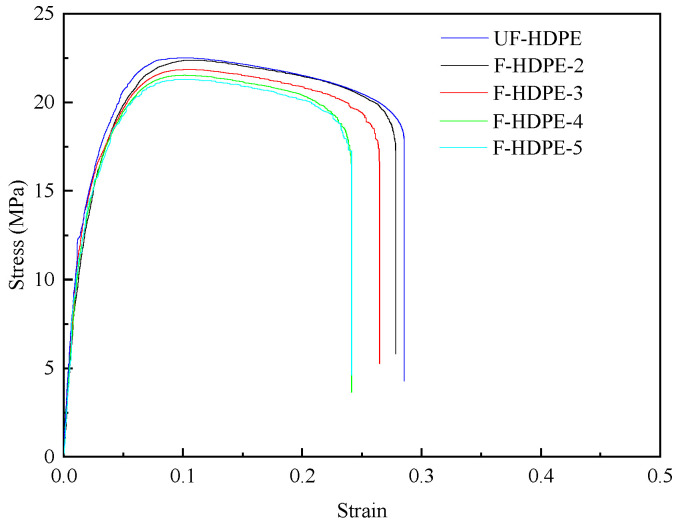
The mechanical properties of foamed samples with different injection times.

**Figure 5 polymers-17-01773-f005:**
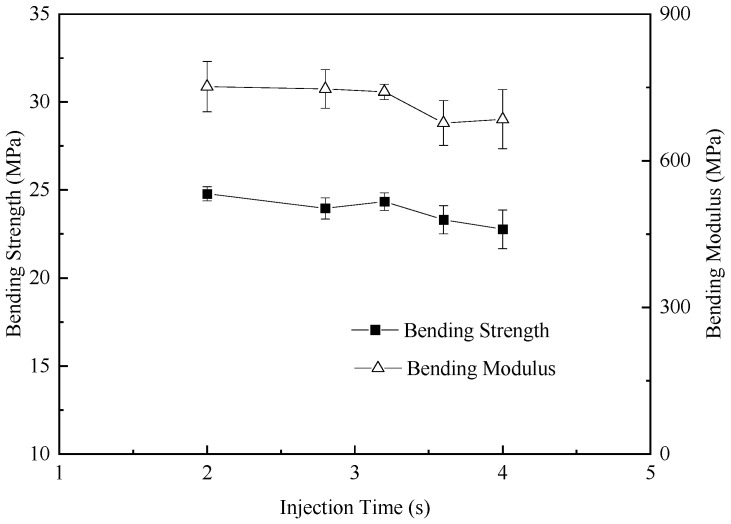
Bending properties of foamed samples with different injection times.

**Figure 6 polymers-17-01773-f006:**
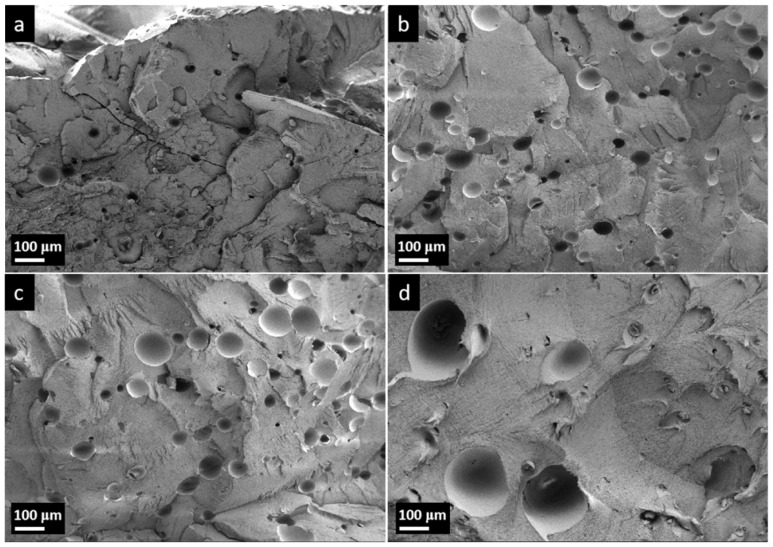
SEM images (Mag = 200) of foamed samples with different nozzle temperatures: (**a**) 190 °C, (**b**) 200 °C, (**c**) 210 °C, and (**d**) 220 °C.

**Figure 7 polymers-17-01773-f007:**
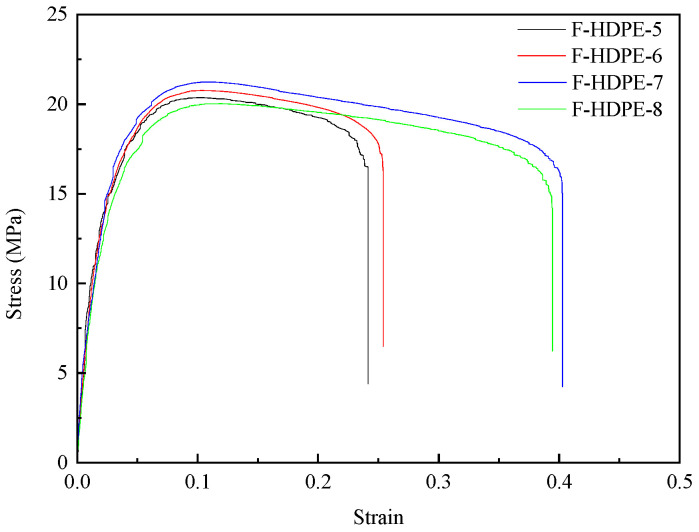
The mechanical properties of foamed samples with different nozzle temperatures.

**Figure 8 polymers-17-01773-f008:**
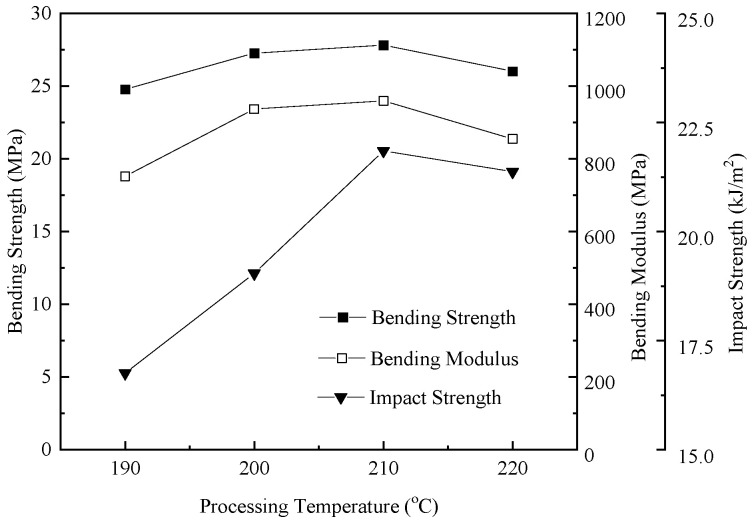
Bending and impact properties of foamed samples with different nozzle temperatures.

**Figure 9 polymers-17-01773-f009:**
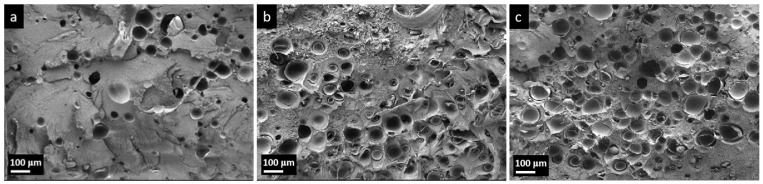
SEM images (Mag = 200) of foamed samples with different TEMs content: (**a**) 1.5 wt.%, (**b**) 3.0 wt.%, and (**c**) 4.5 wt.%.

**Figure 10 polymers-17-01773-f010:**
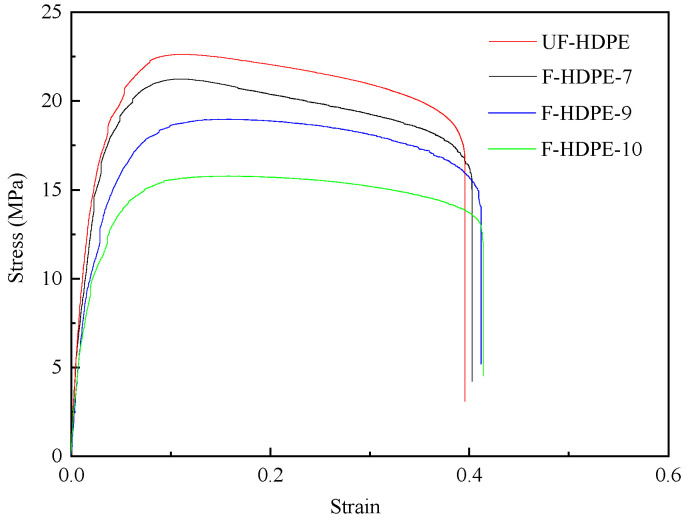
The mechanical properties of foamed samples with different TEM contents.

**Figure 11 polymers-17-01773-f011:**
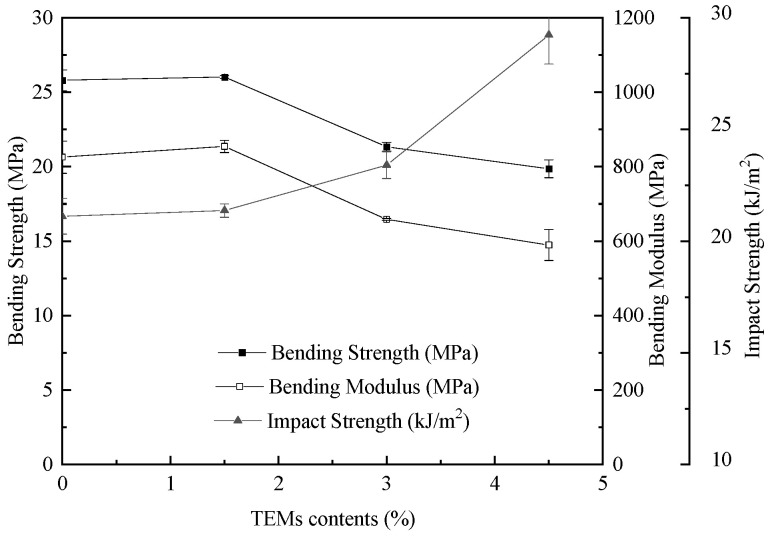
Bending properties of foamed samples with different TEM contents.

**Figure 12 polymers-17-01773-f012:**
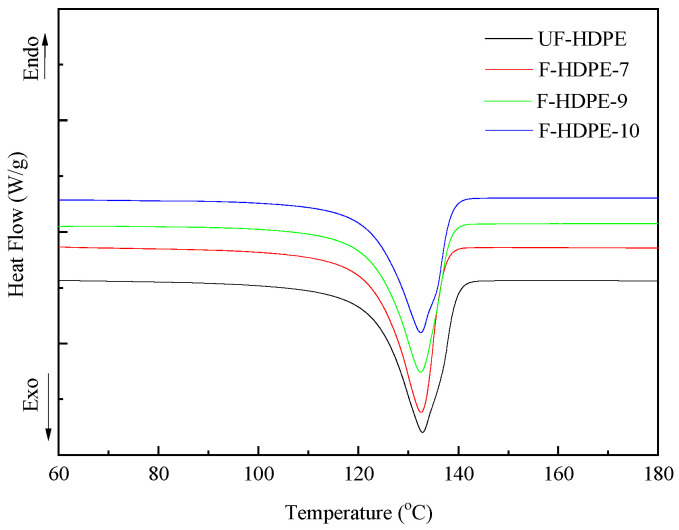
DSC curves of foamed HDPE samples as a function of TEM contents.

**Table 1 polymers-17-01773-t001:** Injection molding parameters of unfoamed and foamed HDPE.

Sample	Nozzle Temperature/°C	Zone 3/°C	Zone 2/°C	Zone 1/°C	Injection Time/s	Content of TMEs/%	Mold Temperature/°C	Injection Pressure/bar	Holding Pressure/bar	Injection Speed/cm^3^·s^−1^	Cooling Time/s
UF-HDPE	210	180	170	160	4.0	0	60	60	20	10	45
F-HDPE-1	190				4.0	1.5			0	60	
F-HDPE-2	190				3.6	1.5					
F-HDPE-3	190				3.2	1.5					
F-HDPE-4	190				2.8	1.5					
F-HDPE-5	190				2.0	1.5					
F-HDPE-6	200				2.0	1.5					
F-HDPE-7	210				2.0	1.5					
F-HDPE-8	220				2.0	1.5					
F-HDPE-9	210				1.0	3.0					
F-HDPE-10	210				1.0	4.5					

**Table 2 polymers-17-01773-t002:** Mechanical properties of foamed HDPE with different injection times.

Sample	Young’s Modulus/MPa	Ultimate Tensile Strength/MPa	Strain at Break/%	Sample Density/g·cm^−3^	Density Reduction/%
F-HDPE-1	1202.9 ± 65.2	22.5 ± 1.9	28.54 ± 1.99	0.9451 ± 0.0091	0.16
F-HDPE-2	1142.0 ± 43.1	22.4 ± 1.8	27.85 ± 1.97	0.9422 ± 0.0047	0.46
F-HDPE-3	1106.0 ± 50.5	22.3 ± 0.5	26.48 ± 1.08	0.9363 ± 0.0092	1.09
F-HDPE-4	1038.3 ± 64.8	21.9 ± 1.2	24.13 ± 1.64	0.9209 ± 0.0028	2.71
F-HDPE-5	935.3 ± 76.2	21.2 ± 0.8	23.03 ± 1.72	0.9066 ± 0.0012	4.23

**Table 3 polymers-17-01773-t003:** Foam density, cell diameter, and cell density results of samples.

Sample	Cell Diameter (μm)	Cell Density (Cells·cm^−3^)
F-HDPE-5	59.53	1.80 × 10^9^
F-HDPE-6	70.42	5.32 × 10^8^
F-HDPE-7	89.72	4.39 × 10^8^
F-HDPE-8	127.44	3.85 × 10^8^
F-HDPE-9	87.43	6.01 × 10^8^
F-HDPE-10	76.70	6.78 × 10^8^

**Table 4 polymers-17-01773-t004:** Mechanical properties of foamed HDPE with different nozzle temperatures.

	Young’s Modulus/MPa	Ultimate Tensile Strength/MPa	Strain at Break/%	Sample Density/g·cm^−3^	Density Reduction/%
F-HDPE-5	935.3 ± 76.2	21.2 ± 0.8	23.03 ± 1.72	0.9066 ± 0.0012	4.23
F-HDPE-6	1089.5 ± 83.6	21.7 ± 0.7	25.40 ± 1.92	0.8984 ± 0.0008	5.09
F-HDPE-7	1172.3 ± 67.0	22.6 ± 0.2	39.45 ± 1.77	0.8922 ± 0.0018	5.75
F-HDPE-8	944.7 ± 53.7	19.4 ± 0.3	37.06 ± 1.76	0.8813 ± 0.0012	6.90

**Table 5 polymers-17-01773-t005:** Mechanical properties of foamed HDPE with different TEM contents.

	Young’s Modulus/MPa		Ultimate Tensile Strength/MPa		Strain at Break/%		Density/g·cm^−3^	
		Reduction		Reduction		Reduction		Reduction
UF-HDPE	1330.1 ± 69.2		22.8 ± 1.3		37.06 ± 1.12		0.9466 ± 0.0023	
F-HDPE-7	1172.3 ± 67.0	−11.87%	22.6 ± 0.2	−0.91%	39.45 ± 1.27	6.45%	0.8922 ± 0.0018	−5.75%
F-HDPE-9	910.7 ± 58.8	−31.54%	19.0 ± 1.3	−16.95%	40.87 ± 1.43	10.28%	0.7952 ± 0.0020	−15.99%
F-HDPE-10	839.1 ± 15.6	−36.92%	15.8 ± 0.2	−30.97%	41.39 ± 1.25	11.69%	0.7768 ± 0.0070	−17.94%

## Data Availability

The original contributions presented in this study are included in the article. Further inquiries can be directed to the corresponding author.

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
