# Peer review of "Preparation of Microcellular High-Density Polyethylene with Thermal Expandable Microspheres"

_polymers, 2025, doi:10.3390/polym17131773_

Round 1
Reviewer 1 Report
Comments and Suggestions for Authors
This work investigated the microstructure and mechanical properties of foamed high-density polyethylene (HDPE) using thermal expandable microspheres (TEMs) by injection moulding. Overall, this is a meaningful study on polymer foaming processing, which well match the scope of Polymers. This manuscript shows the detailed date, well characterized materials, accurate English expression and clear ideas and strategies, although it is less originalities. Therefore, I recommend the minor revision for this manuscript. In addition, I still have some questions as following.
- Please double check again your author’s name Yangi Weicheng? ANG Wei-cheng?
- I think your data in mechanical properties is redundant, for example 22.632 MPa, 1172.258 MPa. All of them can only be accurate to one or two decimal places.
- Line 89, the expression e.g. usually needs to be Italic.
- Line 349 Foam HDPE need to be revised to Foamed HDPE.
- Please explain and highlight the advantages of microcellular. It was obvious that the density of TEMs foamed HDPE only had a less than 10 % reduction, which is slight even negligible. In my opinion,
- Have you considered that all of your mechanical tests need to be normalized by density?
Author Response
The details of the responses to reviewer's insightful comments can be found in the attachment.

Reviewer 2 Report
Comments and Suggestions for Authors
The article should be carefully revised. At the beginning, the authors write that they use three blowing agents and inject samples from them (FHDPE from 1 to 10, so the research plan should include 31 items), then they examine the blowing agents themselves and focus only on the PG31 blowing agent. It should be clearly explained both in the text and with photos and graphs which blowing agent the results refer to.
Line 103 - why was HDPE with such a low MFR used? Isn't it a version for extrusion?
Table 1. There are more basic injection parameters, the most important are: temperatures (injection temperature, mold temperature), pressures (holding pressure, injection pressure), times (injection time, holding time, cooling time), injection speed. Holding pressure plays a significant role in the case of moldings with a blowing agent. It is even possible to abandon the holding phase. What were the values ​​of these parameters?
Table 1. Why were these particular parameter combinations adopted? Was the porophore added in mass or volume?
Did the authors compare the SEM structure in the core of the molded part with the structure of the skin? This could be an interesting topic for future research.
Error bars should be added to graphs and tables showing strength properties.
For SEM images, a magnification should be added in the description of the drawing.
Citations should probably be marked with Arabic numerals.
Author Response

(The authors gave the same response as above.)

Round 2
Reviewer 2 Report
Comments and Suggestions for Authors
The revised article is, in my opinion, suitable for publication. The authors have clarified the remarks in the Cover letter.
Regards